# The Impact of an Anchoring Layer on the Formation of Tethered Bilayer Lipid Membranes on Silver Substrates

**DOI:** 10.3390/molecules26226878

**Published:** 2021-11-15

**Authors:** Indrė Aleknavičienė, Martynas Talaikis, Rima Budvytyte, Gintaras Valincius

**Affiliations:** Institute of Biochemistry, Life Sciences Center, Vilnius University, LT-10257 Vilnius, Lithuania; Indre.aleknaviciene@gmc.vu.lt (I.A.); martynas.talaikis@gmc.vu.lt (M.T.); gintaras.valincius@gmc.vu.lt (G.V.)

**Keywords:** tethered bilayer lipid membranes, self-assembled monolayers, silver films, molecular anchors, electrochemical impedance spectroscopy, surface-enhanced Raman scattering

## Abstract

Tethered bilayer lipid membranes (tBLMs) have been known as stable and versatile experimental platforms for protein–membrane interaction studies. In this work, the assembly of functional tBLMs on silver substrates and the effect of the molecular chain-length of backfiller molecules on their properties were investigated. The following backfillers 3-mercapto-1-propanol (3M1P), 4-mercapto-1-butanol (4M1B), 6-mercapto-1-hexanol (6M1H), and 9-mercapto-1-nonanol (9M1N) mixed with the molecular anchor WC14 (20-tetradecyloxy-3,6,9,12,15,18,22 heptaoxahexatricontane-1-thiol) were used to form self-assembled monolayers (SAMs) on silver, which influenced a fusion of multilamellar vesicles and the formation of tBLMs. Spectroscopic analysis by SERS and RAIRS has shown that by using different-length backfiller molecules, it is possible to control WC14 anchor molecules orientation on the surface. An introduction of increasingly longer surface backfillers in the mixed SAM may be related to the increasing SAMs molecular order and more vertical orientation of WC14 at both the hydrophilic ethylenoxide segment and the hydrophobic lipid bilayer anchoring alkane chains. Since no clustering of WC14 alkane chains, which is deleterious for tBLM integrity, was observed on dry samples, the suitability of mixed-component SAMs for subsequent tBLM formation was further interrogated by electrochemical impedance spectroscopy (EIS). EIS showed the arrangement of well-insulating tBLMs if 3M1P was used as a backfiller. An increase in the length of the backfiller led to increased defectiveness of tBLMs. Despite variable defectiveness, all tBLMs responded to the pore-forming cholesterol-dependent cytolysin, vaginolysin in a manner consistent with the functional reconstitution of the toxin into phospholipid bilayer. This experiment demonstrates the biological relevance of tBLMs assembled on silver surfaces and indicates their utility as biosensing elements for the detection of pore-forming toxins in liquid samples.

## 1. Introduction

The development of biological membrane models emerged over the past few decades to systematically study fundamental processes at lipid bilayer interfaces, such as the membrane proteins functioning, cell–cell signaling, or protein–membrane interactions. Biomimetic membrane models should provide a simplified system for coherent investigation of the membrane while maintaining the fundamental membrane characteristics, such as membrane fluidity or electrical sealing properties [1]. Tethered bilayer lipid membranes (tBLMs) are considered as a comprehensive experimental platform for membrane biosensors. They have been used in various studies ranging from the analysis of biological membrane structure and functions, studies of the membrane–protein and cell–membrane interactions, antigen and antibody binding, also applications as biosensors and energy-generating devices [2]. In such biomembrane models, a fluidic lipid bilayer is attached and stabilized on a solid surface via a thin organic layer. 

Self-assembled monolayers (SAMs) are widely used to tether bilayer lipid membranes to the surface. Most often, binary mixed SAMs consisting of long-chain anchors and short strand backfiller molecules are used [3,4,5,6,7]. The terminal lipide-like aliphatic motif of anchor molecules penetrates the bilayer of phospholipids and attaches it to the planar support by linking to a surface via chemical interaction. For this, the anchor molecules contain chemically reactive terminal groups that bond covalently to a surface. In the case of noble metals, thiols or disulfides are usually used as chemically active groups [8]. Short-chain molecules, backfillers, are frequently used to dilute molecular anchors on the surface, providing the required space for an internal submembrane reservoir [7] and biologically relevant fluidity of the proximal to a surface phospholipid layer of the tethered membrane. Similar chemistry is used to link backfillers to a surface. The size of backfillers and density of long-chain anchors determine the physical properties of such submembrane reservoirs, which is essential for the lipid membrane functionality and protein incorporation [9,10,11]. Attachment of tBLMs to noble metals allows monitoring of the biologically relevant events with surface-sensitive techniques, including surface plasmon resonance spectroscopy (SPR), measurements with a quartz-crystal microbalance (QCM), surface-enhanced Raman scattering (SERS), and electrochemical techniques, such as electrochemical impedance spectroscopy (EIS) [12,13].

Despite the significant interest in tBLMs, there have only been a few reports on a systematic investigation on the effect of the molecular composition and structure of the supporting/tethering assemblies on tBLM formation [14,15,16,17,18]. Most recent studies are focused on the effect of compositions on anchor molecules [4,9,10,18,19]. The impact of short-chain backfiller molecules on tBLM properties is poorly investigated, and only a few papers on the subject can be found. In particular, Liedberg and co-workers analyzed and described the mixed SAMs with variable surface densities of the anchors before and after adsorption of phospholipids employing ellipsometry, contact angle goniometry, and reflection−absorption infrared spectroscopy (RAIRS) [20]. Yet another study described the submembrane arrangement as well as the role of water in shaping the molecular architecture of the tethered bilayer [21].

As was shown in several previous works, the properties of the submembrane space separating the solid surface from the bilayer play an important role in determining the EIS response of tBLMs [22]. The overall measured conductivity of tBLMs is dependent on the chemical composition of the submembrane reservoir and is dominated by it, but not the conductivity of channels or defects in the membrane [3,23]. In our previous study, we developed a novel experimental platform to form tethered bilayer membranes on silver surfaces (Figure 1). It was demonstrated that longer molecular chain thiols, such as 3-mercapto-1-propanol (3M1P), can serve as surface backfillers, successfully eliminating the corrosion of the silver surface [24]. The objective of the current work was to establish the effects of different backfillers along with the molecular anchor WC14 [4,11,25], which were used to form phospholipid bilayer tethering SAMs on the silver surfaces. The WC14 and similar molecular compounds have already been established as functional anchors to form highly insulating tBLMs [4,5,21,24]. Four different backfillers were studied: 3-mercapto-1-propanol (3M1P), 4-mercapto-1-butanol (4M1B), 6-mercapto-1-hexanol (6M1H), and 9-mercapto-1-nonanol (9M1N) (Figure 2). SERS and RAIRS were applied to analyze conformational changes in the membrane-anchoring mixed monolayer. The electrical properties of SAMs and tBLMs formed with different backfillers were determined by EIS. We also explored the interactions of different mixed SAMs with lipids and confirmed an efficient and biologically relevant tBLM formation via small multi-lamellar vesicle fusion by EIS which is suitable for studying the effects of proteins (peptides) to the membranes. The biological relevance was evaluated by studying the reconstitution of pore-forming toxin-vaginolysin in silver-supported tBLM, containing different length-chain backfiller molecules.

## 2. Results and Discussion

### 2.1. Spectroscopic Characterisations of Anchoring SAMs

It is known that the composition and structural properties of SAMs, which are used to anchor phospholipid bilayers to surfaces, affect the functional properties of the tethered bilayer lipid membranes [5]. In this work, the interactions between components of self-assembled monolayers deposited from mixture solutions of long-chain WC14 and short-chain backfillers (mixed SAMs) were explored by electrochemical reductive desorption. From the desorption curves of a single component SAMs (colored lines in Figure 3), it can be seen that the length of the carbon chain in the backfiller affects the Ag–S cleavage potential and the onset of the detachment of SAM molecules. The desorption potential depends on (a) the metal–thiolate bond strength, (b) the magnitude of lateral interactions between the surface molecules, and (c) the interactions between surface molecules and solution ions [26]. For Au(111)-adsorbed alkanethiols terminated with –CH_3_ and –COOH, desorption potential shifts by −15 mV with each additional CH_2_ unit in the molecule due to lateral interactions [27]. Here an increase in the backfiller carbon number from 3 to 6 pushed the desorption potential maxima from −1.02 to −1.17 V. Addition of 20 mol% of WC14 to the bathing mixture increased the resultant SAM stability, which is attested by the shift of the desorption maxima to more negative potentials (grey lines in Figure 3). Such a shift indicates that backfiller and anchor molecules interacted by mutually stabilizing each other in a mixed monolayer. However, the cathodic shift remained constant at about −50 mV for all but 9M1N/WC14 mixed SAMs. This suggests that the strength of interactions between the lipid-like anchors and the backfillers is independent of the carbon chain length of the backfiller molecules. It is noteworthy that the intensity of the desorption peak correlated with the amount of surface adsorbed molecules. As the molecules occupy energetically different surface domains, the desorption curve may contain several peaks, or the peaks may differ in broadness. For the studied system, the nonzero intensity beyond the first desorption peak (to the negative portion of the potential window) is also attributed to the more strongly bound molecules to the surface, therefore rendering it difficult to inspect the relative amount of surface adsorbed molecules.

To gain a deeper understanding of anchoring monolayer structure in the electrochemical double layer, the surface vibrational spectroscopy measurements were carried out for Ag-adsorbed SAMs. To probe the state of the headgroup and the structure of the alkyl chains, surface-enhanced Raman spectroscopy (SERS) was employed (Figure 4). The conformer structure of the S–C motif of the headgroup was recognized from the gauche ν(C–S)_G_ and trans ν(C–S)_T_ vibrational modes typically found in 600–750 cm^–1^ range. The two conformers are depicted in Figure 4C. It should be noted that molecules contribute to the C–S stretching modes according to their surface concentration, therefore for the mixed monolayers (80:20% for backfiller/WC14), most of the spectral intensity was expected from the short-chain backfillers. The ν(C–S)_G_ wavenumbers downshifted from 632 cm^–1^ by 16 cm^–1^ indicating the reduction in the C–S force constant with increasing carbon number. On the other hand, no significant shift was observed for the ν(C–S)_T_ at 696 cm^–1^.

Only for the shortest molecule tested, the 3M1P, the gauche conformational mode (632 cm^–1^) was of considerable intensity compared to trans (696 cm^–1^) because of relatively small attractive van der Waals interactions among carbon chains. Trans and gauche conformations were considered for the molecules, which adopted vertical and more flat-lying configurations with the surface, respectively (Figure 4C). 

As the chain length increased, the gauche band sharply diminished. The more quantitative analysis was given by the trans-to-gauche ratio (T/G) calculated from the integrated intensities, Aν(C–S)_T_/Aν(C–S)_G_. The T/G increased from 2.8 for 3M1P to 18.2 for 9M1N in one-component monolayers and from 1.9 to 17.2 for that of mixed SAMs (Figure 4D). For 100% WC14, the calculated 11.5 T/G ratio was less than that for 6M1H and 9M1N, presumably due to the relatively bulky hydrophobic part composed of two alkane chains that give enough room for conformational flexibility in the S–C and ethylene oxide (EO) parts of the molecule. The higher T/G ratio in longer backfiller SAMs may be related to increasing molecular order and more vertical orientation of backfillers on the surface, which may effectively shrink the submembrane reservoir volume when incorporated into tBLM architecture. 

Being specific mostly for the backfiller, SERS give little information on the long-chain anchoring molecules. Specific for WC14, all-trans ν(C–C)_T_ spectral mode at 1129 cm^–1^ was only observed for the WC14 monolayer composed from the 100% anchor concentration. This band was previously shown to be diagnostic of water and electric potential induced rearrangement of hydrophobic alkane chains into molecular surface clusters [13,21]. The clustering effect was linked with the formation of defected lipid tBLM membranes [5,13]. In the present study, the absence of the ν(C–C)_T_ spectral mode suggests that no clustering takes part in the formation of mixed SAMs or during the SERS measurements, indicating the principal suitability of mixed-component SAMs to be employed in a subsequent tBLM formation.

Contrary to SERS, the RAIRS spectra of mixed SAMs presented in Figure 5 were dominated by the WC14 spectral lines. Considering the area under the curve in the C–H stretching range (2750–3050 cm^–1^), mixed SAMs had on average five times the area of the corresponding single component SAMs (solid and dashed lines in Figure 5). Therefore, regardless of lower concentration, the major contribution to the mixed monolayers came from WC14. The most intense broad feature near 1133 cm^−1^ in the 100% WC14 spectrum was assigned to the asymmetric stretching of the EO fragment ν_as_(C–O–C) [4,28]. Twisting of CH_2_ of the EO fragment was found at 1250–1252 cm^–1^, wagging and scissoring at 1347–1351 and 1460–1464 cm^–1^, respectively [28]. It is noteworthy that WC14 can adopt several conformations, including helical structure with repeated trans–gauche–trans internal rotations around the bonds in the (–CH_2_–CH_2_–O)_6_ segment [29]. The distribution of different EO conformations was revealed by a broad ν_as_(C–O–C) mode that decomposed into several highly overlapped elements. The precise positions of the EO components were determined from the analysis of the second derivatives of the RAIRS spectra. For the 100% WC14 monolayer, the positions were found at 1114, 1133, and 1151 cm^–1^. The lowest energy component at 1114 cm^–1^ was related to highly ordered crystalline helical structure [4,30], while the higher frequency modes were assigned to amorphous structure [28]. Several other spectral modes were found to indicate both the ordered (949 and 1349 cm^–1^) and unordered (1326, 1249, and 1036 cm^–1^) EO structures [28,29].

Owing to the surface selection rule, only the vibrations with a nonzero component of transition dipole moment (TDM) to surface normal were spectroscopically allowed. A sharp increase in the intensity of ν_as_(C–O–C) mode revealed that EO becomes more vertically oriented when contacting longer backfiller molecules. In other words, the more vertical arrangement of backfiller molecules pushes the EO segment into adopting vertical orientation. Such an effect was the most pronounced with C9 and had the least effect with C3 and C4 due to different degrees of lateral interactions between molecules of the same and different natures, respectively. The second derivative RAIRS spectra clearly showed that the position of the lower frequency component upshifted from 1109 to 1115 cm^–1^, while the modes at 1136 and 1151 cm^–1^ became increasingly stronger for EO when in contact with 9M1N. Supporting that, the CH_2_ (EO) wagging mode 1347 cm^–1^ upshifted by 4 cm^–1^, indicating the melting of crystalline-like EO structure [28,31]. These data indicate that not only longer backfiller molecules orient EO segments more vertically with the surface, but also increase the surface fraction of WC14 molecules, which have melted structure in EO.

The C–H stretching region was composed of symmetric ν_s_(CH_2_) and asymmetric ν_as_(CH_2_) stretching modes of methylene groups at 2856 and 2917–2928 cm^–1^ and WC14 terminal methyl group modes ν_s_(CH_3_) and ν_as_(CH_3_) at 2878 and 2965 cm^–1^, respectively. There has already been discussion of these modes being largely from the alkyl chains of the WC14 rather than the EO segment, [4,31] allowing the analysis of the conformational order of the hydrophobic WC14 chains independent of EO. Highly ordered crystalline Ag-adsorbed alkanes have their ν_s_ and ν_as_ methylene vibrations at around 2847 and 2916 cm^–1^ [32]. A departure to the higher wavenumbers from those positions attests to the increase in structural flexibility and gauche defects. Regardless of the backfiller, WC14 alkane chains appeared to be melted as determined from the spectral positions of ν_s_ and ν_as_ of methylene groups. For the mixed SAMs, the disorder in hydrophobic chains could be easily ascribed to the low surface concentration of long-chain molecules.

Yet more valuable information can be extracted from the intensities of ν(CH_3_) and ν(CH_2_) modes. The TDM for methyl and methylene lay in the direction of C–CH_3_ and perpendicular to the C–C chain axis, respectively. Therefore the analysis of their relative intensities provided a measure for the collective orientation of alkane chains. Because of the complex C–H stretching region, we refrained from the decomposition of the experimental spectra into components. However, it was quite clear that the I[ν_as_(CH_3_)]/I[ν_as_(CH_2_)] ratio changed from ~1/3 to ~2/3 when 3M1P was replaced with 9M1N. Thus, the TDM for the methyl group became more vertical with the surface. These data indicate that not only does the EO chain in WC14 tilt closer to the surface normal with an introduction of increasingly longer surface backfiller molecules in the mixed SAM, but also the hydrophobic alkane chains that are situated further from the surface (Figure 5C,D).

### 2.2. EIS Analysis of Mixed SAM‘s 

The electrochemical impedance spectroscopy (EIS) was used to evaluate the structural and functional properties of anchoring monolayers deposited using a mixture solution of WC14 (20%, mol) and the following backfillers (80%, mol): 3-mercapto-1-propanol (3M1P), 4-mercapto-1-butanol (4M1B), 6-mercapto-1-hexanol (6M1H), and 9-mercapto-1-nonanol (9M1N). EIS response from those SAMs is shown in Figure 6. The spectra are presented in the complex capacitance plots (Figure 6). They exhibited a semi-circular shape, consistent with the capacitive behavior of a near ideally insulating dielectric layer. The radius of a semicircle was proportional to the electrical capacitance of the layer. The largest radius was observed for the 3M1P backfiller layer, indicating the highest capacitance value C_SAM_ of about 6 μF/cm^2^ (Figure 6), which did not differ significantly from those of the 30% mixed HC18/βME SAMs used to accomplish tBLMs via vesicle solvent exchange procedure [4]. An increasing molecular length of a backfiller was followed by the decrease in SAM capacitances. The mixed SAM containing 9M1N backfiller exhibited the lowest capacitance SAMs of ∼2.2 μF/cm^2^. C_SAM_ values differed significantly from the ones of 100% WC14 SAM [7], suggesting that anchor molecules in mixed SAMs were laterally distributed on the surface due to the presence of the backfillers. The observed nearly linear variation of SAM capacitances vs. inverse length of the backfiller suggested a similar structural arrangement of these components in the range of compounds from 4M1B to 9M1N. 

The EIS data complement and confirm the spectroscopic analysis (Figure 3, Figure 4 and Figure 5) performed. Decreasing capacitance values (from 6 μF/cm^2^ with 3M1P to ~2.2 μF/cm^2^ with 9M1N) by prolonging the backfiller correlated with spectroscopic data, which showed an increasing molecular order and more vertical orientation of backfillers on the surface. Moreover, SERS analysis of dry samples showed the absence of the ν(C–C)_T_ spectral mode of WC14 molecules in mixed SAMs spectra. This suggests that no clustering of WC14 in alkane chains takes part in the formation of mixed SAMs [5]. Such data indicate the principal suitability of mixed-component SAMs to be employed in a subsequent tBLM formation. The mixed SAMs with variable backfillers: 3M1P, 4M1B, 6M1H, and 9M1N were further used for the assembly of tBLMs via the vesicle fusion process.

### 2.3. EIS Analysis of tBLMs 

As described earlier, tBLMs spontaneously form upon exposure of anchoring mixed SAMs on silver substrates to vesicle solutions [25]. The assembling strategies and physical properties of tBLM were mostly dependent on the backfiller molecules and the type of anchor and the nature of the substrate. The physical properties of the tBLM could be controlled by a mixed SAM WC14/backfiller ratio, which determined the rigidity of the bilayer [7]. Densely packed tether lipids at the surface (no backfilling with spacer) typically lead to very high membrane resistance; however, this can be a problem to the functional reconstitution of protein [33].

While the SERS measurements indicated the mixed SAMs formation on the roughened Ag surface, the EIS was applied for investigations of electrical properties of tBLMs formed on the silver substrate using mixed SAMs containing a variety of backfillers. To compare the propensity of the different molecular lengths of backfillers to form tBLMs, a series of vesicle fusion experiments with DOPC/Chol (ratio 60:40%) were carried out. Figure 7 displays EIS spectra in Cole–Cole and Bode plots of tBLM, created by vesicle fusion on a functionalized silver surface, containing different backfillers: 3M1P, 4M1B, 6M1H, and 9M1N (Figure 7A,B). EIS features in all cases suggested the formation of the lipid bilayer after vesicle fusion. Upon membrane formation, all tBLMs with different backfillers showed proportional spectral changes in EIS with the formation of the phospholipid bilayer.

In a previous study, we reported the ability of WC14/3M1P SAMs to form intact tBLMs by vesicle fusion on silver support [24]. In this study, in all cases, the semi-circular part of the EIS spectra sharply decreased (Figure 7B compared to Figure 6). After vesicle fusion, the high-frequency semi-circular arch now pointed to all values spanning around ~0.6 μF/cm^2^, not depending on the SAM backfiller. Even the capacitances of mixed SAMs, containing 6M1H and 9M1N that were below 4 μF/cm^2^ (Figure 6), have decreased after vesicle fusion suggesting the formation of the lipid bilayer. The full Cole–Cole plot exhibited a typical “two semicircle” shape, which is a characteristic of disrupted integrity of tBLM (Figure 7B). From the mathematical analysis by Valincius et.al., such a well-expressed secondary semicircle in Cole–Cole EIS plots signal the higher number of natural defects occurring in the initial membrane [34].

In parallel, the Bode plots exhibited a minimum of the admittance phase spectra and a step-like feature of the admittance modulus spectra, which appeared at the same frequency point f_min_, below 1 Hz in the case of 3M1P. Those two significant points moved towards higher frequencies as the molecular length of backfiller increased in sequence: 3M1P, 4M1B, 6M1H, and 9M1N (Figure 7A lower panel). This EI spectral feature indicates an impaired isolation property of tBLM in the case of longer backfillers, also suggesting a heterogeneous defect distribution and higher defect number upon tBLM formation as predicted by theoretical analysis of EIS [35].

Variation of EIS spectral features of tBLMs in the backfiller sequence: 3M1P, 4M1B, 6M1H, and 9M1N seems unexpected for several reasons. First, increasing the length of the backfiller molecules decreased Helmholtz capacitance (C_H_) which, according to analysis in [22], induced a small shift of f_min_ towards higher frequencies. However, such shifts were marginal, and as was shown in [22], a 3-fold decrease in C_H_ should shift f_min_ from 15 to 31 Hz. In our case, the observed shift spanned more than two orders of a magnitude from 1 to 100 Hz. Second, one may expect that increasing the molecular length of a backfiller should impair surface mobility of anchor molecules WC14 which tend to form surface clusters upon contact with water milieu [5]. Cluster formation tends to increase defectiveness of tBLMs, thus causing a strong shift of f_min_ toward higher frequencies [5]. Thus, inhibition of cluster formation should decrease f_min_, which we observed the opposite effect. 

Because the position of f_min_ reflected the defectiveness of tBLMs we may assume that in the sequence of backfillers 3M1P, 4M1B, 6M1H, and 9M1N, there were some underlying factors that caused the defectiveness of tBLMs to increase. Such variable factor may be the surface concentration of molecular anchors WC14, which was shown in [7], strongly affects the defectiveness of tBLMs. Quite likely, changing the backfiller in the solution from which mixed SAM was made, resulted in changes of the surface concentration of the molecular anchor WC14, even though concentrations of SAM components in the solution were constant. Such a variation may be induced by variation in hydrophobicity of the backfillers from relatively low in case of 3M1P to high in case of 9M1N. The increasing hydrophobicity caused the increase in the chemical potential of backfiller, thus facilitating more hydrophobic components to form bonds on the silver surfaces, as was observed in similar experiments on gold [36]. Consequently, the surface concentration of 9M1N was likely higher compared to the one of 3M1P causing an increased defectiveness in case of the former.

### 2.4. A Biological Relevance of tBLM on Silver Substrates Containing Different Short-Chain Backfiller Molecules

To evaluate possible applicability of silver-supported tBLMs as a phospholipid biosensor platform, the bilayer lipid membrane containing DOPC/Chol (molar ratio 60:40%) was formed on mixed SAMs WC14/backfiller (in ratio 20:80%), containing different length-chain backfiller molecules and reconstitution of membrane-inserting protein was carried out. 

The cholesterol-containing tBLMs were affected with a cholesterol-dependent cytolysin vaginolysin (VLY) produced by the Gram-negative bacteria *Gardnerella vaginalis*, which acts on the membrane only in the presence of cholesterol. When bound to lipid bilayer VLY oligomerizes and disrupts the bilayer lipid membrane by the formation of a pore [37].

Figure 8 displays EIS Bode plots of the tBLMs after interaction with 2 nM of VLY depending on incubation time. The step-like feature in the admittance modulus curve dropped down after interaction with VLY, signaling an increase in the conductance of tBLMs (Figure 8 (upper panel)). During the 30 min incubation, the value of admittance modulus in the f_min_ point increased nearly two-fold. Additionally, on an admittance phase vs. frequency plot, minima developed, which shifted towards a high-frequency edge as the incubation time of VLY increased (Figure 8 (lower panel)). The continuous phase shift in EIS spectra attests for the increasing damage of tBLMs by the pore-forming toxin with time, as observed in tBLMs assembled on gold supports [37]. Other reasons than transmembrane pore-formation that possibly may affect tBLM stability were excluded by tests involving the neutralizing antibodies, which exhibited full protection of tBLMs in the presence of VLY [37].

Despite the inferior initial electrical properties of tBLMs on SAMs composed using longer backfillers (4M1B, 6M1H, and 9M1N), the impact of VLY was still clearly visible in spectral changes (see Appendix A). Such data suggest that despite significantly higher defectiveness of tBLM the sensitivity to pore-forming proteins can still be observed for tBLMs assembled using longer chain backfillers. Because the concentration of SAM components on the surface may differ from the one in SAM forming solution [36] it is conceivable that the increase in the backfiller molecular chain length should be followed by the decrease in WC14 concentration. Such action would decrease the surface density of anchoring molecules WC14 thus providing worse insulating properties of tBLMs. In conclusion, we state that the anchor SAM, containing WC14 and 3M1P (20:80 mol%) as a backfiller may be used as a biosensing molecular element to detect activities of the pore-forming toxins implicated in bacterial infections.

## 3. Materials and Methods

### 3.1. Fabrication of Silver SERS Sensors

As previously described [38], SERS substrates were made of microscope glass slides in dimensions of 25 m × 75 m × 1 m from ThermoFisher Scientific (London, UK). The slides were prescribed in advance with a diamond wheel into 30 equal parts (15 × 2). Amorphous SERS active nanostructures were fabricated using a micro-machining system FemtoMaster (ELAS, UAB, Vilnius, Lithuania), equipped with a femtosecond laser PHAROS (Light Conversion, UAB, Vilnius, Lithuania) working at the second harmonics at 515 nm. The position of the samples was controlled using precision positioning stages (Aerotech Inc. Pittsburgh, PA, USA), which were synchronized with the pulse picker to deliver 500 pulses/mm, allowing up to 150 mm/s fabrication speed to be reached. Fabrication was carried out in the air, removing most ablated particles from the working zone with a particle exhaust system. All devices were controlled by automation software (Direct Machining Control, Vilnius, Lithuania). Before the vacuum deposition of silver, nanostructured glass slides were washed in 2-propanol and blow-dried with a filtered (5 µm filter) nitrogen stream. A metal film of silver ca. 170 nm thick was deposited by magnetron sputtering using the PVD75 system (Kurt J. Lesker Co., Pittsburg, PA, USA), working in a direct current (DC) regime under the real-time quartz microbalance control.

### 3.2. Deposition of Silver Film 

Microscopic glass slides in dimensions of 25 m × 75 m × 1 m from ThermoFisher Scientific (UK) were used as a substrate for desorption, RAIRS, and EIS experiments, respectively. Before the vacuum deposition of metals, the substrates were sonicated in propanol. After washing in sulfuric acid, rinsed thoroughly with a generous amount of deionized (18.2 Ω·cm^2^, Mili-Q, Millipore, Burlington, MA, USA) water and dried down under a stream of nitrogen (99.99%). The metal films of chromium ± 3 nm and silver ± 120 nm were deposited by the magnetron sputtering using the PVD75 system (Kurt J. Lesker Co., Pittsburg, PA, USA) under the real-time quartz microbalance control. Sputtering parameters for 2-inch diameter metal targets were as follows: Cr-power 200 W, sputtering current 0.50 A at 4.5 mTorr argon pressure; Ag-power 120 W, sputtering current 0.26 A at 5.5 mTorr argon pressure. Before coating, the deposition chamber was evacuated to <10^−6^ mTorr to the residual pressure. Ultrahigh purity, scientific grade argon (AGA, Malmö, Sweden) was used for the plasma sputtering.

### 3.3. Self-Assembled Monolayers Preparation

Freshly prepared SERS sensors and flat silver electrodes were immersed in 0.1mM (total thiol concentration) of ethanolic solution mixture containing the tether WC14 (20–tetradecyloxy-3,6,9,12,15,18,22 heptaoxahexatricontane–1–thiol; synthesized in–house) [7] and four different short-strand backfiller molecules at molar ratio 20:80 mol% (unless advised otherwise): 3-mercapto-1-propanol (3M1P), 4-mercapto-1-butanol (4M1B), 6 mercapto-1-hexanol (6M1H), and 9-mercapto-1-nonanol (9M1N) (Sigma–Aldrich, St. Louis, MO, USA) (Figure 2). Incubation of the silver substrates in the solutions was carried out for 3 h to form self-assembled monolayers (SAMs). Then the substrates were washed in pure ethanol and dried under the stream of nitrogen.

### 3.4. Tethered Bilayer Lipid Membrane Preparation

Samples with the SAMs were assembled into a 14 well electrochemical cell, and each of the wells was exposed to a multilamellar vesicles solution for up to 30 min, during which tBLMs were formed. Vesicle suspensions for tBLM formation were prepared from 1,2–di-oleoyl–sn–glycero–3–phosphocholine (DOPC) and cholesterol from Avanti Polar Lipids Inc. (USA) at a molar ratio of 60:40 mol% in a 0.01 M chloride-free phosphate buffer solution containing NaH_2_PO_4_ and Na_2_HPO_4_ (pH 5.4), total lipid concentration—1 mM. tBLM formation by the fusion of multilamellar vesicles is described in detail elsewhere [39]. Before adding the toxin Vaginolysin, the tBLMs were washed, and the electrochemical cell was filled with the 0.05 mM NaClO_4_ phosphate buffer solution (pH = 7).

### 3.5. Electrochemical Reductive Desorption

Reductive desorption was performed for a single component, and mixed SAMs adsorbed at Ag surface in deaerated 0.5 M NaOH + 0.1 M Na_2_SO_4_ aqueous solution at a sweep rate of −50 mV/s using AutoLab PG101 (Methorms, The Netherlands) potentiostat. Experiments were conducted in a three-electrode system with the Ag/AgCl electrode as a reference.

### 3.6. SERS Measurements

For SERS measurements, we employed a LabRam HR800 (Horiba Jobin Yvon) Raman microscope equipped with a thermoelectrically cooled (−90 °C) CCD camera (DU920P-BR-DD) and 600 lines/mm grating. Spectra were excited with a CW 785 nm laser radiation. The laser power was restricted to 2 mW at the sample and focused using a 10×/0.25 objective lens (Olympus). The overall integration time was 100 s. Three independent measurements were taken to calculate mean intensity ± SD. 

### 3.7. Reflection-Adsorption Infrared Spectroscopy (RAIRS)

RAIRS spectra were collected using Vertex 80v spectrometer (Bruker, Inc., Ettlingen, Germany) equipped with liquid nitrogen-cooled MCT narrow band detector and the RAIRS accessory reflecting p-polarized light at grazing 80° angle. The spectral resolution was set to 4 cm^−1^, an aperture to 3.5 mm, the spectra were acquired by averaging 500 interferogram scans. The spectrometer and sample chamber were evacuated (~2 mbar) during the measurements. The spectrum of a silver-adsorbed hexane-d_13_-thiol monolayer was used as a reference. To determine the precise positions of spectral bands, the fitting experimental spectrum with Gaussian/Lorentzian shape components and analysis of the second derivative spectrum was carried out.

### 3.8. EIS Measurements

Electrochemical impedance spectroscopy measurements were performed in 0.05 mM NaClO_4_ phosphate buffer solution (pH = 7) using Zennium electrochemical workstation (Zahner GmbH, Karlsruhe, Germany). The EI spectra were recorded in a potentiostatic mode with a 10-mV alternating current perturbation voltage at 0 V vs. Ag|AgCl|NaCl (aq. Sat.) with a potential 0 mV respective to the standard hydrogen electrode while the auxiliary electrode was a platinum wire (99.99% purity, Aldrich; diameter 0.25 mm) coiled around the glass cylinder of the reference electrode. Measurements were carried out in a home-built Teflon electrochemical cell holder, which contained 14 separate vials with the surface area of the working electrode 0.16 cm^2^ in each vial. This allowed us to carry out 14 independent experiments on the same metal-plated glass slide. Measurements were carried out in the frequency range from 0.1 Hz to 100 kHz with 10 logarithmically distributed measurement points per decade unless otherwise indicated. 

## 4. Conclusions

In this work, we explored the effect of different compositions of molecular anchors on the formation of tBLM on silver. Four backfiller molecules (3M1P, 4M1B, 6M1H, 9M1N) were selected to dilute a long strand anchor (WC14) and assemble the functional and biologically relevant tBLMs. The structural properties and the molecular conformation of mixed membrane-anchoring monolayers on silver were determined by SERS and RAIRS spectroscopic techniques. Spectroscopic analysis by SERS and RAIRS showed that using different-length backfiller molecules, it is possible to control an orientation of WC14 anchor molecules on the surface. The higher T/G ratio in longer backfiller SAMs may be related to the increasing molecular order and more vertical coordination of backfillers. Moreover, the ethylene-oxide (EO) chain in WC14 tilts closer to the surface normal with increasingly longer surface backfiller molecules in the mixed SAMs, and the hydrophobic alkane chains are also arranged more upright. Since no clustering effect of WC14 alkane chains was observed on dry samples, the mixed-component SAMs were successfully employed in subsequent tBLM formation and further investigated by the electrochemical impedance spectroscopy (EIS). 

We showed that mixed SAMs on silver is a suitable substrate for the spontaneous fusion of multilamellar vesicles that lead to the formation of tBLMs. No adverse corrosion effects, such as described in [24], were observed. The molecular length of backfiller affects the dielectric properties of tBLM and their functional properties as ionic insulators. In particular, we found that the defectiveness of tBLMs increases in the following sequence of backfillers: 3M1P, 4M1B, 6M1H, and 9M1N. We related such effect to a variation in the surface concentration of backfillers due to their different hydrophobicity, an effect analogous to the one described earlier [36]. Due to the high flexibility of anchor molecules, the WC14 alkane chains can possibly interact with backfiller molecules, especially the longer ones, which may also be the reason for the higher tBLM defectiveness. The tBLMs formed on silver substrates were found to be biologically relevant. All compositions responded to vaginolysin, one of the pore-forming toxins, indicating typical variations of EIS spectra following functional reconstitution of proteins into membranes [37]. The membrane damaging effects by VLY were reproducible and time-dependent. It suggests the utility of tBLMs on silver substrates as a proper substitute for gold films in designing biosensors for pore-forming toxins. In conclusion, tBLMs on Ag-coated mixed monolayers can be engineered and optimized as a valuable biologically relevant experimental model for reconstitution of transmembrane proteins and/or for protein–lipid interaction studies.

## Figures and Tables

**Figure 1 molecules-26-06878-f001:**
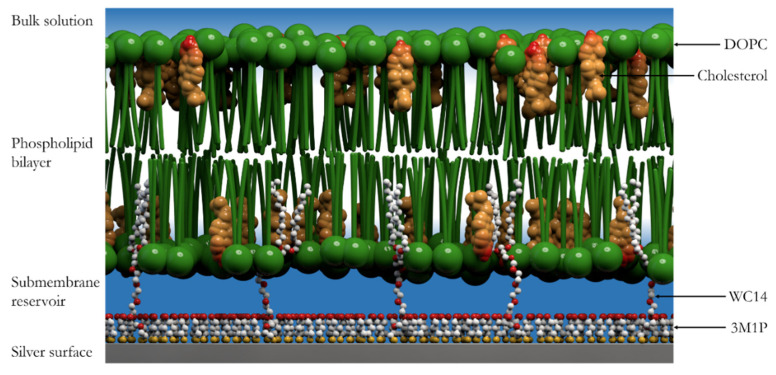
tBLM model on silver surface.

**Figure 2 molecules-26-06878-f002:**
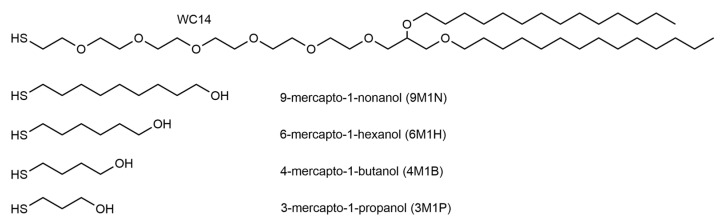
Chemical structure of molecular anchors.

**Figure 3 molecules-26-06878-f003:**
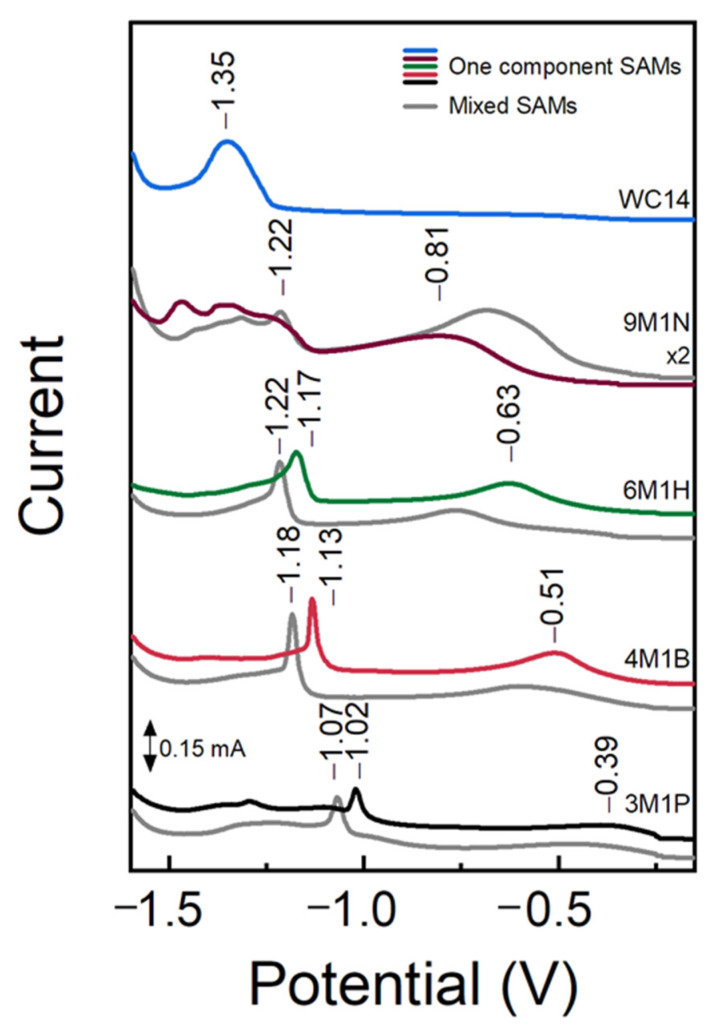
Potentiodynamic curves of one component SAMs (colored lines) and mixed component SAMs (80% backfiller: 20% WC14) (grey lines). Reductive desorption was performed from Ag film in 0.5 M NaOH + 0.1 M Na_2_SO_4_ deaerated aqueous solution at the sweep rate of −50 mV/s. The desorption curves of 9M1N and 9M1N + WC14 SAMs were multiplied by two.

**Figure 4 molecules-26-06878-f004:**
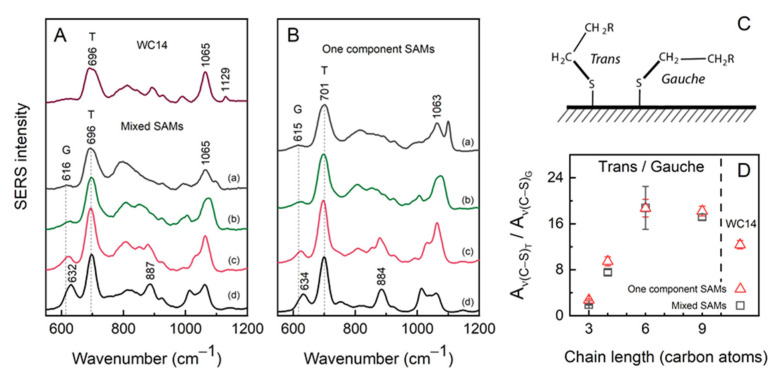
(**A**) SERS spectra of Ag adsorbed WC14 (top curve) and mixed SAMs containing different backfillers and (**B**) one component SAMs. The order of curves is 9M1N (a), 6M1H (b), 4M1B (c), 3M1P (d). Letter G and T denote gauche and trans conformation. (**C**) Schematic representation of surface adsorbed thiol molecules in the gauche and trans conformation. The conformation was defined by the dihedral angle of the –S–C– bonding (**D**) T/G ratio dependence on the chain length in carbon atoms of backfiller. The composition of mixed SAMs was 80% backfiller and 20% WC14. Mean ± SD were calculated from 3 measurements.

**Figure 5 molecules-26-06878-f005:**
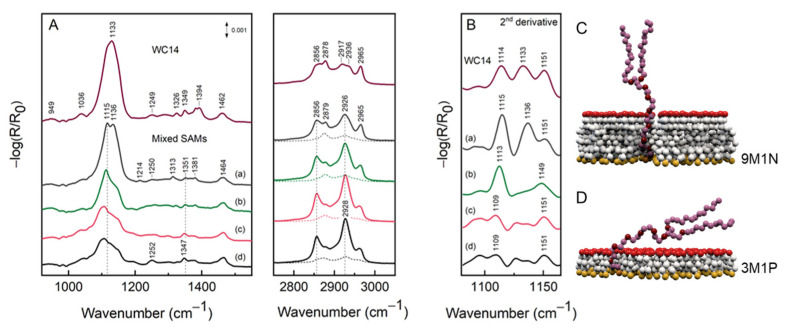
(**A**) RAIRS spectra of Ag adsorbed WC14 (top curve) and mixed SAMs containing different backfillers in the following order: 9M1N (a), 6M1H (b), 4M1B (c), 3M1P, (d) in the 900–1600 and 2750–3050 cm^–1^ range. The composition of mixed SAMs was 80% backfiller and 20% WC14. Dashed curves in C–H stretching region represent one-component monolayers that had, on average, a 5-times lower area under the curve than the corresponding mixed SAMs. (**B**) The second derivative RAIRS spectra in ν_as_(C–O–C) region (multiplied by −1). The arrangement of the spectra corresponded to that of the (**A**) panel. Panels (**C**,**D**)—proposed model of WC14 molecules orientation on Ag surface (at air/solid interface): (**C**) WC14 diluted with 9M1N in ratio 20:80% WC14/3M1P; (**D**) WC14 diluted with 3M1P in ratio 20:80% WC14/9M1N.

**Figure 6 molecules-26-06878-f006:**
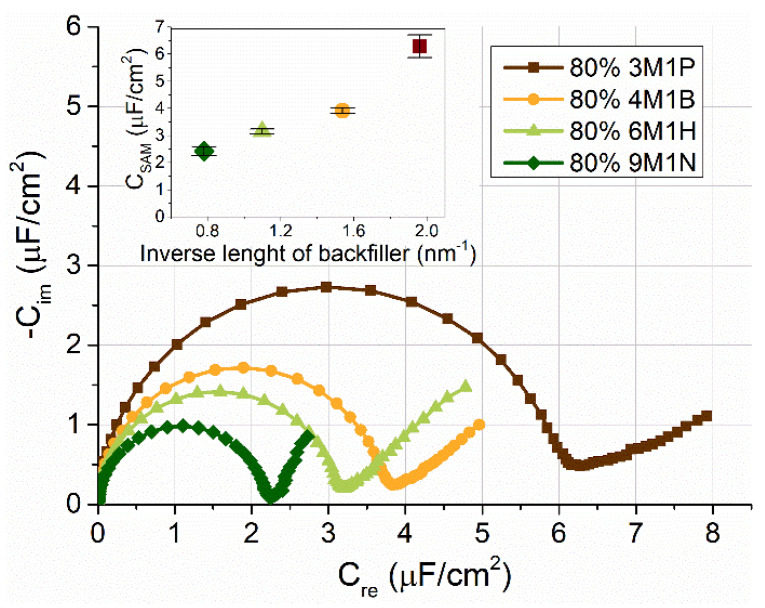
Representative EIS spectra of mixed WC14/backfiller (in ratio 20:80%) SAMs containing different backfillers: 3M1P, 4M1B, 6M1H, and 9M1N. The Cole–Cole complex capacitance plots; Inset demonstrates capacitance values of different-backfiller SAMs as a function of backfiller length (nm^−1^). Frequency range 100 kHz–0.1 Hz.

**Figure 7 molecules-26-06878-f007:**
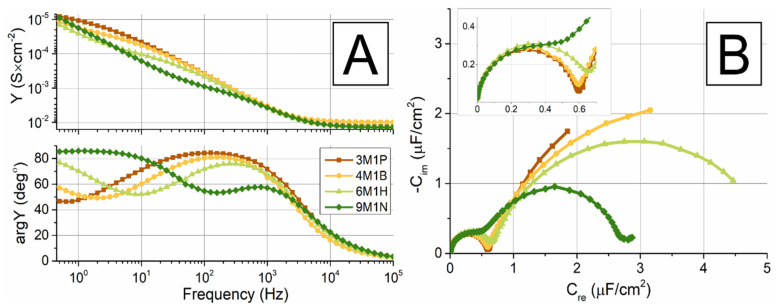
The formation of tBLM on mixed WC14/backfiller SAMs. EIS data of DOPC/Chol (ratio 60:40%) tBLM composed on mixed WC14/backfiller (in ratio 20:80%) SAMs, containing different backfillers: 3M1P, 4M1B, 6M1H, and 9M1N. (**A**) Admittance modulus (upper panel) and phase (lower panel) (Bode) plots (**B**). Capacitance values of tBLMs with different backfillers and WC14 anchors. Inset shows an enlarged area of the plot with a higher frequency (100 kHz to 30 Hz). Full frequency range 100 kHz–0.1 Hz.

**Figure 8 molecules-26-06878-f008:**
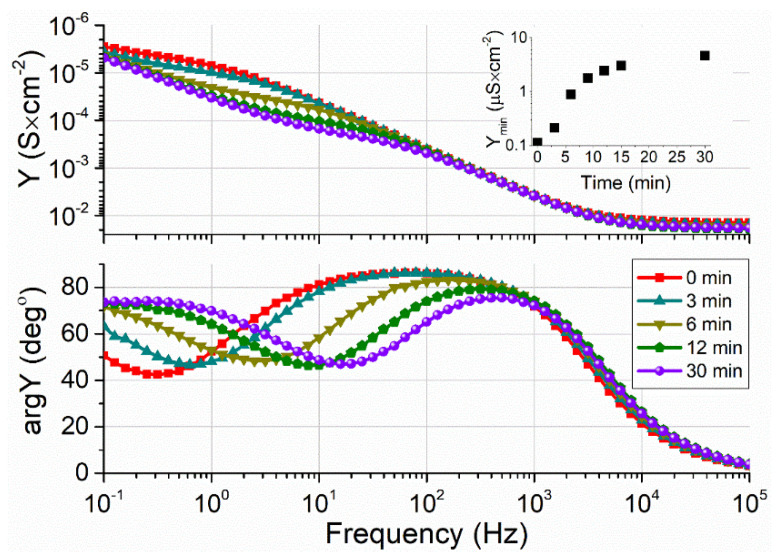
EIS spectral transformations of tBLMs upon exposure to VLY on time (initial, 3 min, 6 min, 9 min, 30 min). Admittance modulus (upper panel) and phase (lower panel) (Bode) plots. Inset shows admittance values at phase minimum (f_min_) point as a function of time. Two-nanometer VLY kinetics on tBLM were formed on mixed WC14/3M1P SAM and completed with DOPC/Chol (in molar ratio 6:4) phospholipid mixture.

## Data Availability

The data presented in this study are available on request from the corresponding authors.

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
