# Peer review of "The Impact of an Anchoring Layer on the Formation of Tethered Bilayer Lipid Membranes on Silver Substrates"

_molecules, 2021, doi:10.3390/molecules26226878_

Round 1

Reviewer 1 Report

Overall, the manuscript is comprehensive and may be accepted. 

Minor issues: 

  1. WC14, SAM in the abstract is not defined.
  2. WC14 is defined only in 432, better be defined when it appears first.
  3. Line 11-14 in the abstract can be made more clear.

Major issues: 

  1. Provide detail about the stability of the tBLMs on silver substrate over time.

  2. How did the author choose 20:80% ratio of WC14/backfiller ? Should this differ if the molar ratio is changed?? Please explain.

  3. It would help understand the density of anchor molecules in the phospholipid structure of tBLM, if authors provide a quantitative comparison (e.g. molar ratio) between tBLM and SAMs with the WC14 anchor molecules and phospholipid bilayer.

  4. To validate further biological relevance of the experiments described herein, a control experiment would have been better, eg. with a non-pore forming event, and/or some experiments to show that “pore-formation” is not due to instability in the structure of the tBLPs.

Author Response

Response to reviewer 1.

Reviewer 2 Report

In the manuscript entitled: “The Impact of Anchoring Layer on Formation of Tethered Bilayer Lipid Membranes on Silver Substrates” Aleknavičienė et al described the effect of different compositions of molecular anchors on the formation of tBLM on silver, revealing the possibility to control an orientation of WC14 anchor molecules on the surface using different length backfiller molecules. This research work uses multiple technologies, EIS, SERS, and RAIRS, to study tethered bilayer lipid membranes, facilitating the exploration of the mechanism of the assembly systems of tBLMs. Overall, this is an interesting work that will advance related biological research. I have a few comments, answering of which may be regarded as a minor revision:

  1. In the paper, the authors mention that there are two conformational forms, trans and gauche conformational modes. Can the authors show these two conformations in either 2D or 3D figures? This will be very helpful to readers especially for people who are not studying this system.
  2. In figure 3, the 9M1N (one component SAMs) does not show any peak. What is the reason for this? And is there any relation between the peak intensities with each backfiller?
  3. In lines 142-144, can the authors show how the 3M1P assembled in gauche and trans conformational modes?
  4. Why did the authors use WC14 to study the assemble system? Based on the authors' conclusions, if using a longer molecular anchor with 9M1N as the backfiller to study, will be the longer molecular anchor be less vertical? Also, is it possible the fatty acid on WC14 can insert into the backfiller layer since the molecules are very flexible?
  5. Can authors give a better explanation for figure 7 including the content in the article and the figure caption? In figure 7B, the authors show the zoomed-in view of figure 7B. Why does 9M1N show the difference between 9M1N and other different backfillers?
  6. In this paper, the authors mixed the molecular anchors and backfillers with a ratio of 20:80. Why did the authors choose this ratio instead of other ratios? What will happen if a higher or lower molecular anchor to backfillers ratio is used?

Author Response

Response to reviewer 2.
